# Selective Brain Damage: Measuring the Disparate Impact of Model Pruning

## Abstract

Neural network pruning techniques have demonstrated it is possible to remove the majority of weights in a network with surprisingly little degradation to top-1 test set accuracy. However, this measure of performance conceals significant differences in how different classes and images are impacted by pruning. We find that certain examples, which we term *pruning identified exemplars (PIEs)*, and classes are systematically more impacted by the introduction of sparsity. Removing PIE images from the test-set greatly improves top-1 accuracy for *both* sparse and non-sparse models. These hard-to-generalize-to images tend to be mislabelled, of lower image quality, entail abstract representations, atypical examples or require fine-grained classification.

## 1 Introduction

Between infancy and adulthood, the number of synapses in our brain first grow and then fall. Synaptic pruning improves efficiency by removing neurons that are redundant and strengthening synaptic connections that are useful for the environment (Rakic et al., 1994). Despite losing 50% of all synapses between age two and ten, the brain continues to function (Kolb & Whishaw, 2009; Sowell et al., 2004). "Use it or lose it" is frequently used to describe the environmental influence of the learning process on synaptic pruning, however there is little scientific consensus on *what* exactly is lost (Casey et al., 2000).

In this work, we ask what is *lost* when we prune a deep neural network. Work on pruning deep neural networks has demonstrated a remarkable ability to sparsify a model to a fraction of the original weights while giving up minimal test-set accuracy (Cun et al., 1990; Hassibi et al., 1993b; Han et al., 2015; Ullrich et al., 2017; Liu et al., 2017; Louizos et al., 2017; Collins & Kohli, 2014; Weigend et al., 1991; Nowlan & Hinton, 1992; Lee et al., 2018b). Gale et al. (2019) show it is possible to prune 90% of all weights in a ResNet-50 network (He et al., 2015) trained on ImageNet (Deng et al., 2009) and only lose less than 3% absolute in top-1 test set accuracy. The ability to prune networks with seemingly so little degradation to accuracy is puzzling. In this work, we address the following questions:

- *Are certain types of examples or classes disproportionately affected by pruning?*
- *How does pruning impact robustness such as sensitivity to image corruptions (blur, noise, contrast) and adversarial examples?*

Answers for these question can provide intuition into the role of additional capacity in deep neural networks and, perhaps more important, provide a principled framework for articulating the trade-offs incurred by compressing deep neural networks. Many of the most promising use cases for compressed models occur in sensitive domains, such as improving access to health care by using machine learning driven diagnostics on mobile phones (Esteva et al., 2017). Pruned or compressed models are frequently favored for deploying deep neural networks onto devices because reducing the number of network weights lowers energy consumption, memory footprint, and latency (Reagen et al., 2016; Chen et al., 2016; Theis et al., 2018; Kalchbrenner et al., 2018; Valin & Skoglund, 2018). For tasks where incorrect predictions can harm human welfare, it is critical that we understand when a machine learning model is qualified to make decisions on real world inputs. To our knowledge, this is the first work to shed light on the trade-offs pruning incurs by considering new measures beyond test accuracy.

The primary findings of our work can be summarized as follows:

1. *In some settings, pruning has a non-uniform impact across classes; a fraction of classes are disproportionately and systematically impacted by the introduction of sparsity. .*

2. *The examples most impacted by pruning, which we term Pruning Identified Exemplars (PIEs), are more challenging for both sparse and non-sparse models to classify.*

3. *Pruning makes deep neural networks less robust to natural adversarial examples and common image perturbations (such as blur, fog and contrast).*

For (1) and (2), we establish consistent findings for different standard architectures on CIFAR-10 (Krizhevsky, 2009) and ImageNet (Deng et al., 2009). We identify *Pruning Identified Exemplars* (PIEs) as images where the modal label differs between a set of sparse and non-sparse models. We find that removing PIEs from the test-set improves top-1 accuracy for *both* a fully parameterized non-sparse model as well as a sparse model (Figure 4).

Toward finding (3), we measure changes to model sensitivity to both common image corruptions and natural adversarial examples using two open source robustness benchmarks: ImageNet-C (Hendrycks & Dietterich, 2019) and ImageNet-A (Hendrycks et al., 2019).We find that both sparse and non-sparse models are brittle and sensitive to ImageNet-A and C, and that this brittleness is amplified at highers level of sparsity (Table 7).

Our findings provide important insights about when pruned models are qualified to make decisions on real world inputs. Our PIE methodology identifies a tractable subset of images which are more challenging for sparse and non-sparse models. Our work suggest that PIE could be useful for tasks where it is important to choose not to classify certain examples when the model is uncertain (Bartlett & Wegkamp, 2008; Cortes et al., 2016b;a; Cortes et al., 2017), to aid interpretability as a case based reasoning tool to explain model behavior (Kim et al., 2016; Gurumoorthy et al., 2017; Caruana, 2000; Hooker et al., 2018) or to surface atypical examples for further human inspection (Leibig et al., 2017).

## 2 A COMPARISON OF SPARSE AND DENSE MODELS

We consider a supervised classification problem where a deep neural network is trained to approximate the function $F$ that maps an input variable $X$ to an output variable $Y$, formally $F : X \mapsto Y$. The model is trained on a training set of $N$ images $\mathcal{D} = \{(x_i, y_i)\}_{i=1}^{N}$, and at test time makes a prediction $y_i^*$ for each image in the test set. The true labels $y_i$ are each assumed to be one of $C$ classes, such that $y_i = [1, ...., C]$.

A reasonable response to our desire for more compact representations is to simply train a network with fewer weights. However, as of yet, starting out with a compact dense model has not yielded competitive test-set performance. Instead, research has centered on training strategies where models are initialized with "excess capacity" which is then subsequently removed through a pruning process. A pruning method $\mathcal{P}$ identifies the subset of weights to remove (i.e. set to zero). A sparse model function, $\hat{F}_t^{\mathcal{P}}$, is one where a fraction $t \in [0, 1]$ of all model weights are set to zero. Equating weight value to zero effectively removes the contribution of a weight as multiplication with inputs no longer contributes to the activation. A non-sparse model function, $\hat{F}_0$, is one where all weights are trainable.

**ImageNet and CIFAR-10 Setup** We consider two classification tasks and models; a wide ResNet model (Zagoruyko & Komodakis, 2016) trained on CIFAR-10 and a ResNet-50 model (He et al., 2015) trained on ImageNet, both using batch normalization (Ioffe & Szegedy, 2015). A key goal of our analysis is to produce findings that are not anecdotal as would be the case when analyzing one trained model of each type. Instead we independently train a population of 30 models for each experimental setting. We train for $32,000$ steps (approximately 90 epochs) on ImageNet with a batch size of 1024 images and $80,000$ steps with a batch size of 128 for CIFAR-10. For ImageNet, the baseline non-sparse model obtains a mean top-1 accuracy of $76.68\%$ and mean top-5 accuracy of $93.25\%$ across 30 models. For CIFAR-10, mean baseline top-1 accuracy is $94.35\%$.

We prune over the course of training to obtain a target end sparsity level $t \in \{0.0, 0.1, 0.3, 0.5, 0.7, 0.9\}$. For example, $t = 0.9$ indicates that $90\%$ of model weights are removed by pruning, leaving a maximum of $10\%$ non-zero weights.

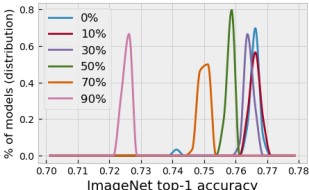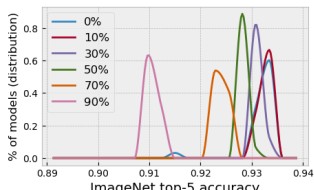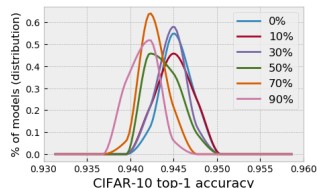

Figure 1: Distributions of top-1 and top-5 model accuracy for populations of independently trained sparse and non-sparse models on ImageNet and CIFAR-10. The distributions for CIFAR-10 top-5 accuracy (not shown) are tightly clustered and overlapping in $[0.9965, 1.0]$. The distributions are fairly tight with one outlier for the ImageNet baseline model.

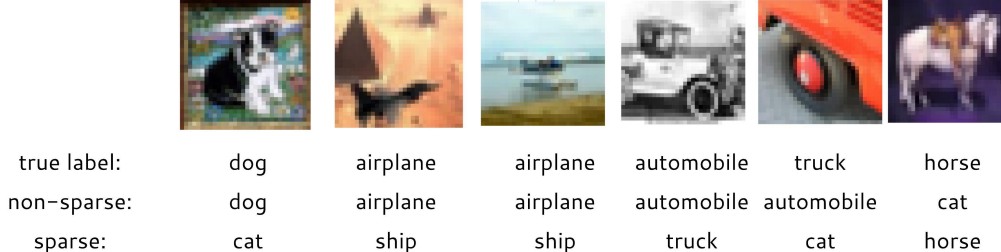

| true label: | dog | airplane | airplane | automobile | truck | horse |
| non-sparse: | dog | airplane | airplane | automobile | automobile | cat |
| sparse: | cat | ship | ship | truck | cat | horse |

Figure 2: Visualization of pruning identified exemplars ($PIE_{30}$) for the CIFAR-10 dataset. This subset of impacted images is identified by considering a set of 30 non-sparse wide ResNet models and 30 models trained to 30% sparsity.

Across all experiments, we use magnitude pruning as proposed by Zhu & Gupta (2017) to identify the weights to remove. Magnitude pruning is a simple rule-based method that thresholds weights at zero that fall below a certain absolute magnitude. It has been shown to outperform more sophisticated Bayesian pruning methods and is considered state-of-the-art across both computer vision and language models (Gale et al., 2019). The choice of magnitude pruning also allowed us to specify and precisely vary the final model sparsity for purposes of our analysis, unlike regularizer approaches that allow the optimization process itself to determine the final level of sparsity (Liu et al., 2017; Louizos et al., 2017; Collins & Kohli, 2014; Wen et al., 2016; Weigend et al., 1991; Nowlan & Hinton, 1992). Although the ability to precisely vary sparsity is required for this experimental framework, we note that our methodology can be extended to other methods. Figure 1 shows the distributions of model accuracy across model populations for the non-sparse and sparse models for ImageNet and CIFAR-10.

## 2.1 CLASS LEVEL PERFORMANCE

Comparisons of the effects of different pruning algorithms or levels of sparsity on classification tasks such as ImageNet have largely centered on top-line metrics such as top-1 or top-5 test-set accuracy averaged across classes, but reliance on these measures hides detail in the model's performance. Comparing only top-1 model accuracy between a baseline and a pruned model amounts to assuming that class accuracies are expected to maintain their relative relationships to the top-1 model accuracy before and after pruning. In this work, we consider whether this is a valid assumption. *Is relative performance unaltered by pruning or are some classes impacted more than others?*

For a given model, we compute the class accuracy $\beta_t^c$ for class $c \in \mathcal{C}$ and sparsity $t \in [0, 1]$. We compute overall model accuracy from the set of class metrics:

$$\beta_t^{\mathcal{M}} = \frac{1}{N_{\mathcal{M}}} \sum_{c \in \mathcal{C}} \beta_t^c * N_c$$

where $N_c$ is the number of examples in class $c$ and $N_{\mathcal{M}}$ is the total number of examples in the data set. If the impact of pruning was uniform, we would expect each class accuracy to shift by the same number of percentage points as the difference in top-1 accuracy between the sparse and non-sparse model. This forms our **null hypothesis** ($H_0$) – the shift in accuracy for class $c$ before and after

pruning is the same as the shift in top-1 accuracy. For each class $c$ we consider whether to reject $H_0$ and accept the **alternate hypothesis** ($H_1$) that pruning disparately affected the class's accuracy in either a positive or negative direction:

$$H_0 : \beta_0^c - \beta_0^{\mathcal{M}} = \beta_t^c - \beta_t^{\mathcal{M}}$$
$$H_1 : \beta_0^c - \beta_0^{\mathcal{M}} \neq \beta_t^c - \beta_t^{\mathcal{M}}$$

Neural net training is most often done in a non-deterministic fashion, and we consider each model $k$ in its population of $K = 30$ models to be a sample of some underlying distribution. Given a class $c$ and a population of $K$ models trained at a sparsity $t$, we construct a set of samples $S_t^c$ of the mean-shifted class accuracy as $S_t^c = \{\beta_{t,k}^c - \beta_{t,k}^{\mathcal{M}}\}_{k=1}^K$.

Evaluating whether the difference between a sample of mean-shifted class accuracy from sparse and non-sparse models is "real" amounts to determining whether two data samples are drawn from the same underlying distribution, which is the subject of a large body of goodness of fit literature (D'Agostino & Stephens, 1986; Anderson & Darling, 1954; Huber-Carol et al., 2002). In this work, we use a two-sample, two-tailed, independent Welch's t-test (Welch, 1947) to determine whether the means of the samples $S_t^c$ and $S_0^c$ differ significantly. If the two samples were drawn from distributions with different means with 95% or greater probability ($p$-value $<= 0.05$), then we reject the null hypothesis and consider the class to be disparately affected by $t$-sparsity pruning relative to the baseline.

After finding the subset of classes for a given $t$-sparsity that show a statistically significant change relative to the baseline, we can quantify the degree of deviation, which we refer to as *normalized recall difference*, by comparing the average $t$-sparse and baseline class accuracies after normalizing for their respective average model accuracies:

$$\frac{1}{K_t} \sum_{k=1}^{K_t} \left( \beta_{t,k}^c - \beta_{t,k}^{\mathcal{M}} \right) - \frac{1}{K_0} \sum_{k=1}^{K_0} \left( \beta_{0,k}^c - \beta_{0,k}^{\mathcal{M}} \right) \tag{1}$$

## 2.2 Pruning Identified Exemplars

How does pruning impact model performance on individual images? A natural extension of the hypothesis testing in the prior section is to consider whether to reject or retain the null hypothesis that the output probability for a given image for a dense and sparse models is equal. However, recent work has highlighted that deep neural networks produce output probabilities that are uncalibrated (Guo et al., 2017; Kendall & Gal, 2017; Lakshminarayanan et al., 2017) and thus cannot be interpreted as a measure of certainty. Deep neural networks do not know what they do not know, and often ascribe high probabilities to out-of-distribution data points or are overly sensitive to adversarially perturbed inputs (Hendrycks & Gimpel, 2016; Nguyen et al., 2014).

We are interested in how model predictive behavior changes through the pruning process. Given the limitations of uncalibrated probabilities in deep neural networks, we focus on the level of disagreement between the predictions of sparse and non-sparse networks on a given image. Let $y_{i,k,t}^*$ be the prediction of the $k$th $t$-sparse model of its population for image $i$ where $t = 0$ denotes an unpruned model, and let $Y_{i,t}^* = \{y_{i,k,t}^*\}_{k=1}^K$ be the set of predictions for the $t$-sparse model population on exemplar $i$. For set $Y_{i,t}^*$ we find the *modal label*, i.e. the class predicted most frequently by the $t$-sparse model population for exemplar $i$, which we denote $y_{i,t}^M$. Exemplar $i$ is classified as $PIE_t$ if and only if the modal label is different between the $t$-sparse model and the unpruned (non-sparse) model:

$$PIE_{i,t} = \begin{cases} 1 & \text{if } y_{i,0}^M \neq y_{i,t}^M \\ 0 & \text{otherwise} \end{cases}$$

We note that there is no constraint that the non-sparse predictions for PIEs match the true label, thus the detection of PIEs is an unsupervised protocol that could in principal be performed at test time.

| Sparsity ($t$) | Model accuracy diff. | Significant | | Largest increase | | Largest decrease | |
|---|---|---|---|---|---|---|---|
| | | # incr. | # decr. | class | norm. diff. | class | norm. diff. |
| 0.1 | -0.0002 | 29 | 22 | toaster | 0.025 | am. chameleon | -0.026 |
| 0.3 | -0.002 | 35 | 34 | bathtub | 0.036 | cleaver | -0.045 |
| 0.5 | -0.008 | 91 | 54 | petri dish | 0.034 | frying pan | -0.047 |
| 0.7 | -0.017 | 189 | 128 | cd player | 0.050 | tow truck | -0.069 |
| 0.9 | -0.041 | 337 | 245 | cd player | 0.088 | muzzle | -0.128 |

Table 1: Summary of class-level results for ImageNet. Only classes passing the significance test are included. The model accuracy difference column reports mean $\beta_t^{\mathcal{M}} - \beta_0^{\mathcal{M}}$ as the percentage point difference between the pruned and baseline model accuracies; a negative value means the pruned model's average accuracy is lower than the baseline model's.

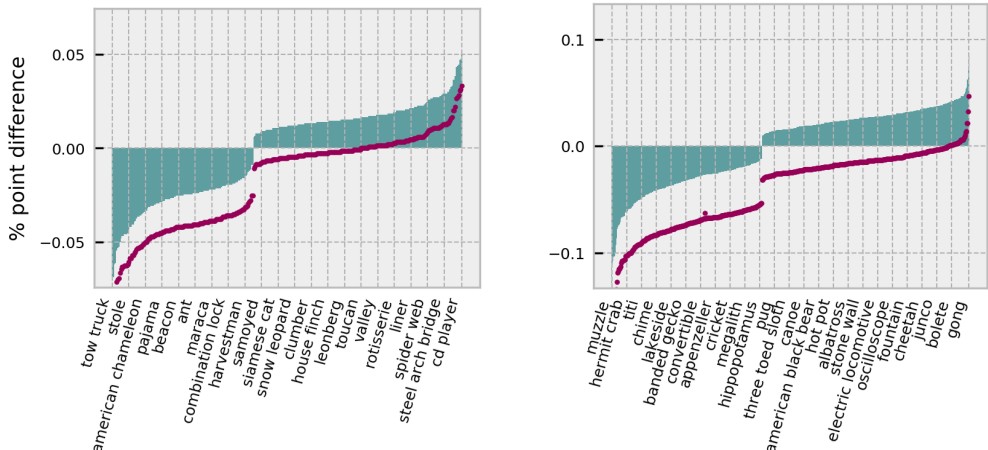

Figure 3: Normalized recall difference (bars) and absolute recall difference (points) per class for 70% sparsity (left) and 90% sparsity (right). The class labels are sampled for readability; there are 317 significant classes for 70% sparsity and 582 significant classes for 90%. Note the difference in scale on the y-axis.

## 3 RESULTS AND DISCUSSION

### 3.1 IMPACT OF SPARSITY ON CLASS LEVEL PERFORMANCE

The impact of magnitude pruning on ImageNet classification is disparate across classes and amplified as sparsity increases. For example, at $10\%$ sparsity only 51 of 1,000 classes in the ImageNet test set exhibit statistically significant change in class accuracy, however at $90\%$ sparsity, accuracy is impacted for 582 classes in a statistically significant way.

The directionality and magnitude of the impact is nuanced and surprising. Our results show that certain classes are relatively robust to the overall degradation experienced by the model whereas others degrade in performance far more than the model itself. This amounts to selective "brain damage" with performance on certain classes evidencing far more sensitivity to the removal of model capacity. Table 1 shows that more classes show a significant *relative* increase in accuracy than a decrease at every level, though the overall model accuracy decreases at every pruning level, indicating that the magnitude of class decreases must be larger in order to pull the model accuracy lower. Figure 3 visualizes the magnitude of the normalized recall differences for 70% and 90% pruning and highlights the degree to which the classes spread from model average.

We performed the same analysis on the CIFAR-10 models and found that while pruning presents a non-uniform impact there are fewer classes that are statistically significant. One class out of ten was significantly impacted at 10%, 30%, and 50%, and two classes were impacted at 90%. We suspect

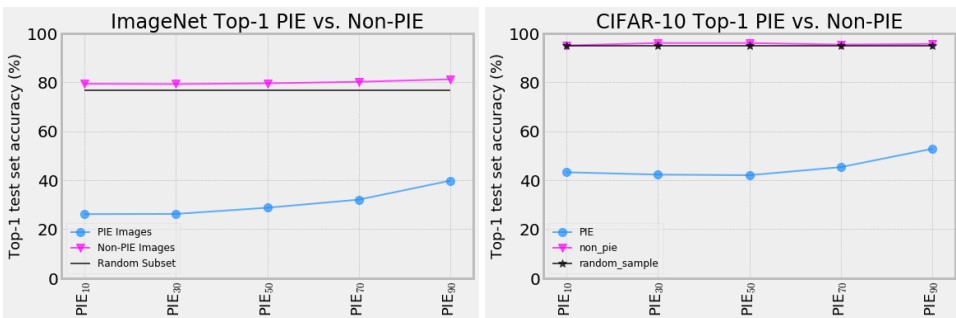

Figure 4: Excluding pruning identified exemplars (PIE) improves test-set top-1 accuracy for both ImageNet and CIFAR-10. This holds for PIE images identified at all levels of sparsity considered. Sparse model find generalizing to PIE more challenging, and inference on PIE images alone substantially degrades generalization performance. **Left:** Average top-1 test-set accuracy across 30 non-sparse ResNet-50 models when inference is restricted to PIE images (blue), non-PIE images (dark purple) and a random sample of the test set (black line) which is a constant independent of PIE sparsity level. **Right:** Average top-1 test-set accuracy across 30 non-sparse wide ResNets trained on CIFAR-10.

that we found less disparate impact for CIFAR-10 because, while the model has less capacity, the number of weights is still sufficient to model the limited number of classes and lower dimension dataset.

## 3.2 IMPACT OF SPARSITY ON INDIVIDUAL EXEMPLARS

At every level of sparsity, for both CIFAR-10 and ImageNet we identify a subset of PIE images. At 10% and 90% sparsity, we classify 3.34% and 10.27% of all ImageNet test-set images respectively as PIEs. For CIFAR-10, PIEs constitute 0.97% and 2.16% of the test set at 10% and 90% sparsity.

**PIEs are more difficult for both sparse and non-sparse models to classify.** In Fig. 4, we compare the test-set performance of a fully parameterized non-sparse model on a fixed number of randomly selected (1) PIE images, (2) non-PIE images and (3) a random sample of the test set. The results are consistent across both CIFAR-10 and ImageNet datasets; removing PIE images from the test-set improves top-1 accuracy for *both* sparse and non-sparse models relative to a random sample. Inference restricted to only PIE images greatly impairs the generalization performance of a model. In the appendix, we include additional plots that show that while all models perform far worse on PIE images, the degradation to performance is amplified as model sparsity increases.

**The most challenging PIEs are identified at low levels of sparsity.** In Figure 4, the lowest test-set accuracy for both sparse and non-sparse models occurs when inference is restricted to PIEs identified at 10% sparsity. Test-set accuracy steadily increases for PIEs identified at higher levels of sparsity. This suggests that sparsity first erodes performance on the images that the model finds the most challenging.

**Why are PIEs impacted more by pruning?** A qualitative inspection of PIEs (Figure 6) suggests that these hard-to-generalize-to images tend to be of lower image quality, mislabelled, entail abstract representations, require fine-grained classification or depict atypical class examples. We conducted a limited human study[1] to inspect a random sample of 400 PIE and non-PIE ImageNet images. We broadly group the properties we codify as indicative of 1) the exemplar being challenging or 2) the task being ill-specified. We introduce these groupings below (after each bucket we report the percentage of PIEs and non-PIEs in each category as a fraction of total PIEs and non-PIE codified):

1. Poorly specified task

---

[1]The authors acted as labelers ahead of the deadline, though the PIE label was shuffled and hidden during labeling. We will reproduce the study using other parties before publication.

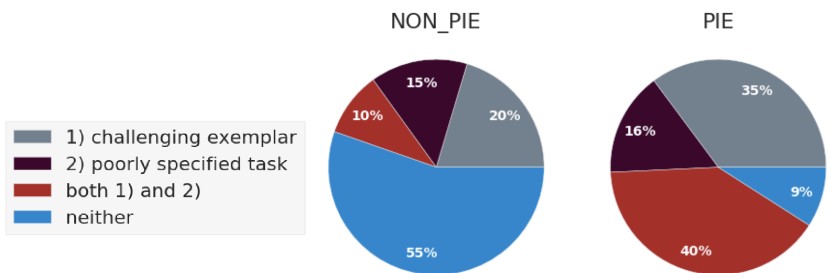

Figure 5: Limited human study of the relative distribution of PIE and non-PIE properties. **Challenging exemplars:** images positively codified as showing *common image corruptions* such as blur or overlaid text, or images where the object is in the form of an *abstract representation* or where the exemplar requires *fine grained classification*. **Poorly specified task:** images where *multiple classes* are visible in the same image, or images with *incorrect or insufficient ground truth.*

- **incorrect ground truth** or **insufficient information** – images where there is not sufficient information for a human to arrive at the correct ground truth label. For example in Fig. 6, the image of the plate of food with the label `restaurant` is cropped such that it is impossible to tell whether the food is in a restaurant or in a different setting. [2.9% of non-PIEs, 9.5% of PIEs]

- **multiple labelled classes are clearly visible in the image** – images depicting multiple objects where a human may consider several labels to be appropriate (e.g., an image which depicts both a `paddle` and `canoe`, `desktop computer` consisting of a `screen`, `mouse` and `monitor`, a `barber chair` in a `barber shop`). [22.3% of non-PIE, 50.8% of PIEs]

2. Challenging Exemplars

- **fine grained classification task** – involves classifying an object that is semantically close to various other classes present the data set (e.g., `rock crab` and `fiddler crab`, `bassinet` or `cradle`, `cuirass` and `breastplate`). [8.3% of non-PIEs, 38.2% of PIEs]

- **exemplar corruptions** – images exhibit common corruptions such as motion blur, contrast, pixelation. We also include in this category images with super-imposed text, an artificial frame and images that are black and white rather than the typical RBG color images in ImageNet. [8.7% of non-PIE, 12.1% of PIE]

- **exemplar abstractions** – the surfaced exemplar depicts a class object in an abstract form such a cartoon, painting, or sculptured incarnation of the object. [1.5% of non-PIE, 3.5% of PIE]

We find that PIEs overindex relative to non-PIEs on certain properties, such as having an *incorrect ground truth label* or *multiple objects*. This suggests that the task itself is often incorrectly specified. Both ImageNet and CIFAR-10 are single image classification tasks, however just over half of the PIEs codified by humans were identified as multi-object images where multiple labels could be considered reasonable (vs. 22.33% of non-PIEs). The over-indexing of incorrectly structured data in PIE hints that the explosive growth in number of parameters in deep neural networks may be solving a problem better addressed in the data cleaning pipeline. However, task mis-specification alone does not fully explain the images where predictive behavior diverges as model capacity is changed. We find that PIE also overindexes on *low quality images* that present corruptions like blur or overlaid text and exemplars that require *fine grained classification*. This suggests that there may be a relationship between number of model parameters and generalization to these types of images. We explore the relationship between capacity and robustness to these types of corruptions more thoroughly in Section 3.3.

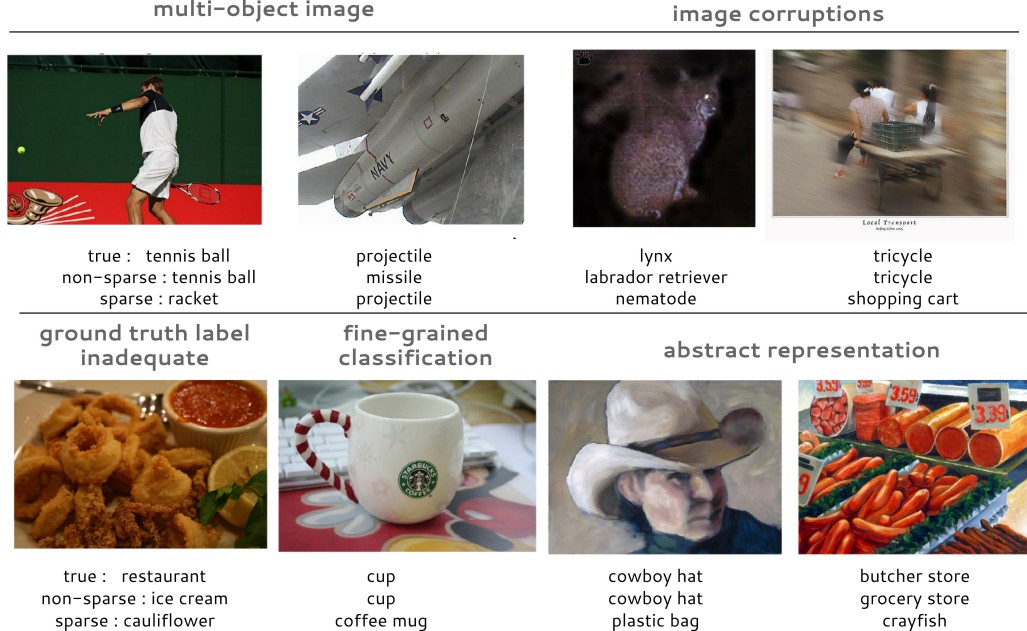

Figure 6: Visualization of $PIE_{90}$: images where modal label differs between 30 non-sparse ResNet-50 models and 30 models trained to $90\%$ sparsity on ImageNet. A qualitative inspection of PIEs suggests that there are certain shared properties between the images most impacted by pruning; these images tend to be of lower image quality and frequently exhibit image corruptions (overlaid text, contrast, motion blur, zoom), mislabelled or unintuitive label, depict class in abstract form, require fine-grained classification or depict atypical class examples.

### 3.3 THE ROLE OF ADDITIONAL CAPACITY

The PIE procedure surfaces exemplars that are harder for both sparse and non-sparse models to classify. Given that PIE surfaces data points where there is the greatest divergence in behavior between sparse and non-sparse models, it is useful to understand the directionality of some of the properties described in the previous section. For example, we noted that PIE frequently surfaces images with corruptions such as motion blur, contrast and pixelation. *Are sparse models more brittle to these type of corruptions?* Many PIEs are often atypical or unusual class examples. We have already noted that model degradation when restricted to inference on PIEs is amplified as sparsity increases. *Does this measure of model brittleness mirror other open source robustness benchmarks?*

**ImageNet-C** ImageNet-C (Hendrycks & Dietterich, 2019) is an open source data set that consists of algorithmically generated corruptions (blur, noise) applied to the ImageNet test-set. We compare top-1 accuracy given inputs with corruptions of different severity. As described by the methodology of Hendrycks & Dietterich (2019), we compute the corruption error for each type of corruption by measuring model performance rate across five corruption severity levels (in our implementation, we normalize the per-corruption error by the performance of the sparse model on the clean ImageNet dataset). ImageNet-C corruption substantially degrades mean top-1 accuracy of non-sparse models Fig. 7. This sensitivity is amplified at high levels of sparsity, where there is a further steep decline in top-1 accuracy. Sensitivity to different corruptions is remarkably varied, with certain corruptions such as gaussian, shot an impulse noise consistently causing more degradation.

**ImageNet-A** ImageNet-A is a curated test set of $7,500$ natural adversarial images designed to produce drastically low test accuracy. We find that the sensitivity of sparse models to ImageNet-A mirrors the patterns of degradation to ImageNet-C and sets of PIEs. As sparsity increase, top-1 and top-5 accuracy further erode, suggesting that sparse models are more brittle to adversarial examples.

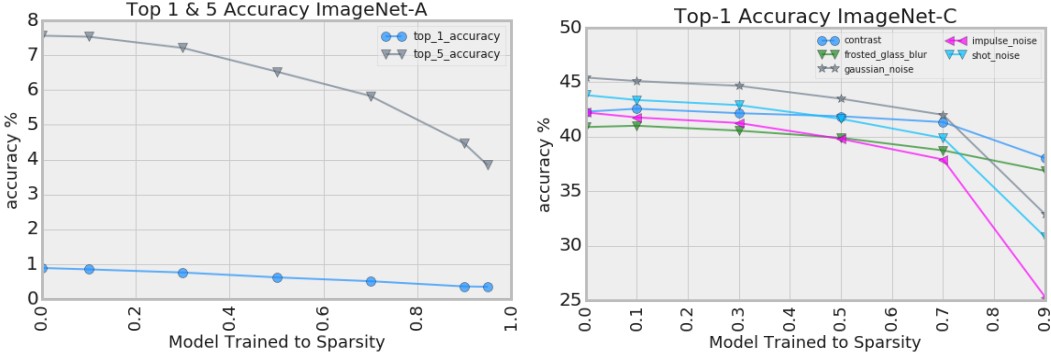

Figure 7: Sparse models are less robust to natural adversarial examples. At high levels of sparsity, models are also more brittle to common image corruptions. We measure the **relative** Top-1 and Top-5 test set ResNet-50 accuracy normalized by average sparse model performance on an uncorrupted ImageNet test set. **Left:** Mean test-set accuracy on ImageNet-A (across 30 models). **Right:** Test-set performance on a subset of ImageNet-C corruptions. An extended list of all corruptions considered is included in the appendix.

## 4 RELATED WORK

Model compression is diverse and includes research directions such as reducing the precision or bit size per model weight (quantization) (Courbariaux et al., 2014; Hubara et al., 2016; Gupta et al., 2015), efforts to start with a network that is more compact with fewer parameters, layers or computations (architecture design) (Howard et al., 2017; Iandola et al., 2016; Kumar et al., 2017) and student networks with fewer parameters that learn from a larger teacher model (model distillation) (Hinton et al., 2015) and finally pruning by setting a subset of weights or filters to zero (Louizos et al., 2017; Wen et al., 2016; Cun et al., 1990; Hassibi et al., 1993a; Strm, 1997; Hassibi et al., 1993b; Zhu & Gupta, 2017; See et al., 2016; Narang et al., 2017). Articulating the trade-offs of compression has overwhelming centered on change to overall accuracy. Our contribution, while limited in scope to model compression techniques that prune deep neural networks, is the first work to our knowledge to consider the limitations of these aggregate performance measures and demonstrate that the impact of pruning in deep neural networks at a class and exemplar level is non-uniform.

We also consider how pruning impacts robustness to natural adversarial examples and image corruptions. We note that recent work by (Hendrycks & Dietterich, 2019; Hendrycks et al., 2019) considers complimentary variant of this question by benchmark ImageNet-A and ImageNet-B robustness across a limited set of dense non-sparse architectures with different numbers of parameters (for example ResNet-50 vs ResNet-101). While our work is focused on understanding the impact of sparsity on an exemplar and class level, one of our key findings is that PIE is far more challenging to classify for both sparse and non-sparse models. Leveraging this subset of data points for interpretability purposes or to cleanup the dataset fits into a broader and non-overlapping body of literature that aims to classify input data points as prototypes – "most typical" examples of a class – ((Carlini et al., 2018; Stock & Cisse, 2017)) or outside of the training distribution (OOD) (Hendrycks & Gimpel, 2016; Lee et al., 2018a; Liang et al., 2018; Lee et al., 2018a; Masana et al., 2018) and work on calibrating deep neural network predictions (Lakshminarayanan et al., 2017; Guo et al., 2017; Kendall & Gal, 2017).

## 5 CONCLUSION

We show that deep neural networks pruned to different levels of sparsity "forget" certain classes and examples more than others. While a subset of classes are systematically impacted, the direction of this impact is surprising and nuanced. Our results show certain classes are relatively impervious to the reduction in model capacity while others bear the brunt of degradation in performance. Pruning identified exemplars are a subset of exemplars where there is a high level of disagreement between sparse and non-sparse models. We show that this subset is universally challenging for models levels at all levels of sparsity to classify and mirrors the sensitivity of sparse models to open source robustness benchmarks ImageNet-C and ImageNet-A.

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

# 6 APPENDIX

## 6.1 MAGNITUDE PRUNING

There are various pruning methodologies that use the absolute value of the weight as way to rank importance and remove from the network weights that are below a user specified threshold. This is often over the course of training; training is punctuated at certain pruning steps and a fraction of weights are set to zero. Many different magnitude pruning methods have been proposed (Collins & Kohli, 2014; Guo et al., 2016; Zhu & Gupta, 2017) that largely differ in whether the weights are removed permanently or can "recover" by still receiving subsequent gradient updates. This would allow certain weights to become non-zero again if pruned incorrectly. While magnitude pruning is often used as a criteria to remove individual weights, it can be adapted to remove entire neurons or filters by extending the ranking criteria to a set of weights and setting the threshold appropriately. Recent work on evolutionary strategies has also leveraged an interative version of magnitude pruning (Mocanu et al., 2018).

In this work, we use the magnitude pruning methodology proposed by Zhu & Gupta (2017). Pruning is introduced over the course of training and removed weights continue to receive gradient updates after being pruned. For ImageNet, each model trains for a total of 32,000 steps. We prune every 500 steps between 1,000 and 9,000 steps. For CIFAR-10, we train the model for 80,000 steps. We prune every 2,000 steps between 1,000 and 20,000 steps. These hyperparameter choices were based upon a limited grid search which suggested that these particular settings minimized degradation to test-set accuracy across all sparsity levels. At the end of training, the final pruned mask is fixed and during inference only the remaining weights contribute to the model prediction.

## 6.2 ADDITIONAL CLASS-LEVEL RESULTS

Tables 2 and 3 provide top-line metrics for ImageNet and CIFAR-10, respectively.

The class-level analysis summary for CIFAR-10 is in Table 6.2. Relative to ImageNet, the percentage of classes significantly impacted at 90% pruning is small: 20% for CIFAR-10 versus 58% for ImageNet. As discussed in the main body, we suspect this is due to CIFAR-10 being a simpler task and the network we started from having much more capacity than necessary for the task.

| Fraction Pruned | Top 1 | Top 5 | # Signif classes | # PIEs |
|:---:|:---:|:---:|:---:|:---:|
| 0 | 76.68 | 93.25 | - | - |
| 0.10 | 76.66 | 93.25 | 51 | 1,694 |
| 0.30 | 76.46 | 93.17 | 69 | 1,819 |
| 0.50 | 75.87 | 92.86 | 145 | 2,193 |
| 0.70 | 75.02 | 92.43 | 317 | 3,073 |
| 0.90 | 72.60 | 91.10 | 582 | 5,136 |

Table 2: ImageNet top-1 and top-5 accuracy at all levels of sparsity, averaged over all runs. The fourth column is the number of classes significantly impacted by pruning.

| Fraction Pruned | Top 1 | # Signif classes | # PIEs |
|:---:|:---:|:---:|:---:|
| 0 | 94.53 | - | - |
| 0.1 | 94.51 | 1 | 97 |
| 0.3 | 94.47 | 1 | 114 |
| 0.5 | 94.39 | 1 | 144 |
| 0.7 | 94.30 | 0 | 137 |
| 0.9 | 94.14 | 2 | 216 |

Table 3: CIFAR-10 top-1 accuracy at all levels of sparsity, averaged over runs. Top-5 accuracy for CIFAR-10 was 99.8% for all levels of sparsity. The fourth column is the number of classes significantly impacted by pruning.

| Sparsity $(t)$ | Model accuracy diff. | Significant | | Largest increase | | Largest decrease | |
|---|---|---|---|---|---|---|---|
| | | # incr. | # decr. | class | norm. diff. | class | norm. diff. |
| 0.1 | -0.0002 | 0 | 1 | - | - | automobile | -0.0016 |
| 0.3 | -0.0006 | 1 | 0 | frog | 0.0022 | - | - |
| 0.5 | -0.0014 | 1 | 0 | truck | 0.0022 | - | - |
| 0.7 | -0.0023 | 0 | 0 | - | - | - | - |
| 0.9 | -0.0039 | 2 | 0 | truck | 0.0030 | - | - |

Table 4: Summary of class-level results for CIFAR-10. Only classes passing the significance test are included. The model accuracy difference column reports mean $\beta_t^{\mathcal{M}} - \beta_0^{\mathcal{M}}$ as the percentage point difference between the pruned and baseline model accuracies; a negative value means the pruned model's average accuracy is lower than the baseline model's. The normalized difference is calculated using Equation 1.

Figure 3 in the main body of the paper visualizes the relative increases and decreases in performance across classes at 70% and 90% pruning on ImageNet. Those figures were shrunk for space; Figure 8 shows the full chart for 70% sparsity for clarity.

### 6.3 Additional PIE Results

In the body of the paper we show the performance of the unpruned model on PIE images found at varying levels of sparsity and showed that performance is worst for $PIE_{10}$ images which we suspect are the most difficult and performance is still poor but better for $PIE_t$ for larger values of $t$. In Figure 10 we plot the performance of the pruned models on PIEs identified at different levels of sparsity and show that the behavior of the pruned models track the behavior of the unpruned model in this regard.

### 6.4 Additional Corruption and Adversarial Results

Sparse models are less robust to natural adversarial examples. At high levels of sparsity, models are also more brittle to common image corruptions. We include the raw ImageNet-C results in Figure 5.

### 6.5 Human Study

A balanced sampled PIE and non-PIE were selected at random and shuffled. The classification as PIE or non-PIE was not known or available in the sample. Questions codified for every image considered:

*Does label 1 accurately label an object in the image? (0/1)*

*Does this image depict a single object? (0/1)*

*Would you consider labels 1,2 and 3 to be semantically very close to each other? (does this image require fine grained classification) (0/1)*

*Do you consider the object in the image to be a typical exemplar for the class indicated by label 1? (0/1)*

*Is the image quality corrupted (some common image corruptions – overlaid text, brightness, contrast, filter, defocus blur, fog, jpeg compression, pixelate, shot noise, zoom blur, black and white vs. rbg)? (0/1)*

*Is the object in the image an abstract representation of the class indicated by label 1? [[an abstract representation is an object in an abstract form, such as a painting, drawing or rendering using a different material.]] (0/1)*

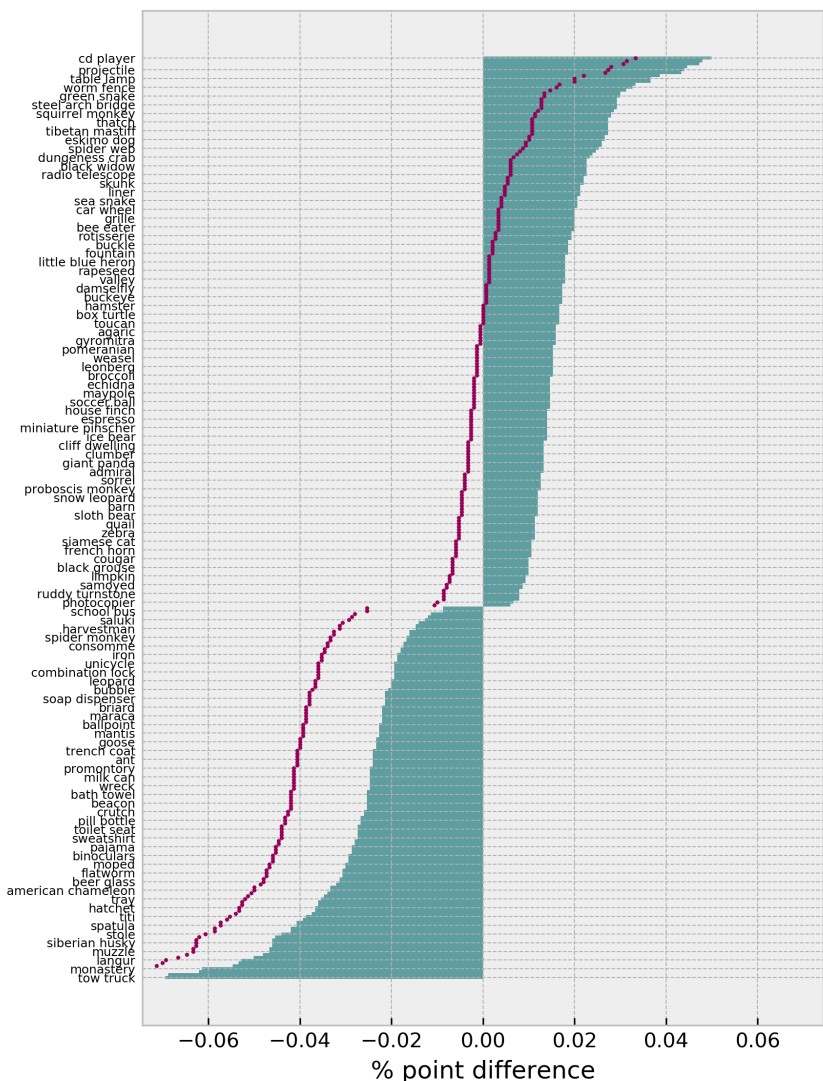

Figure 8: Expanded version of the 70% sparsity chart from Figure 3. Normalized recall difference (bars) and absolute recall difference (points) per class. Every third class label is shown.

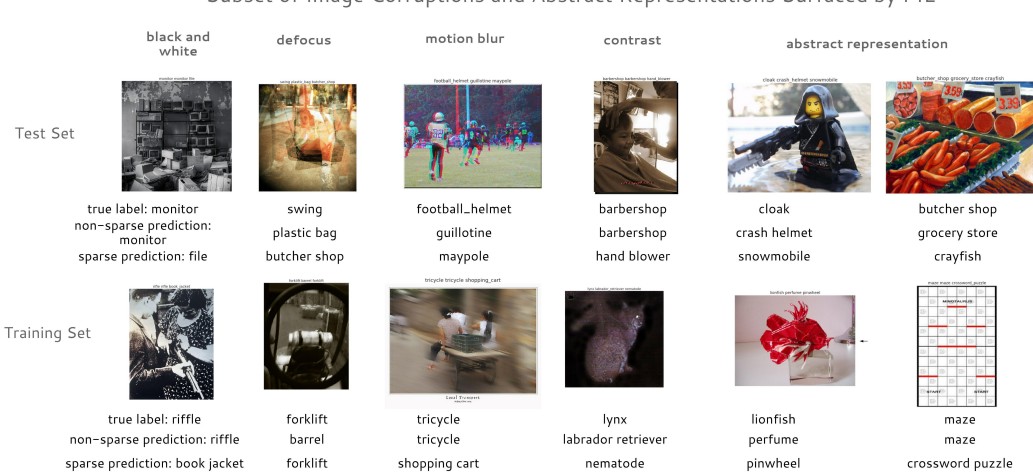

Figure 9: Images surfaced by PIE evidence common corruptions such as motion blur, defocus or post-processing with overlaid text. Many PIE images depict objects in an abstract form, such as a painting, drawing or rendering using a different material. PIEs displayed were identified by comparing the modal label of a set of $90\%$ sparse and non-sparse ResNet-50 models.

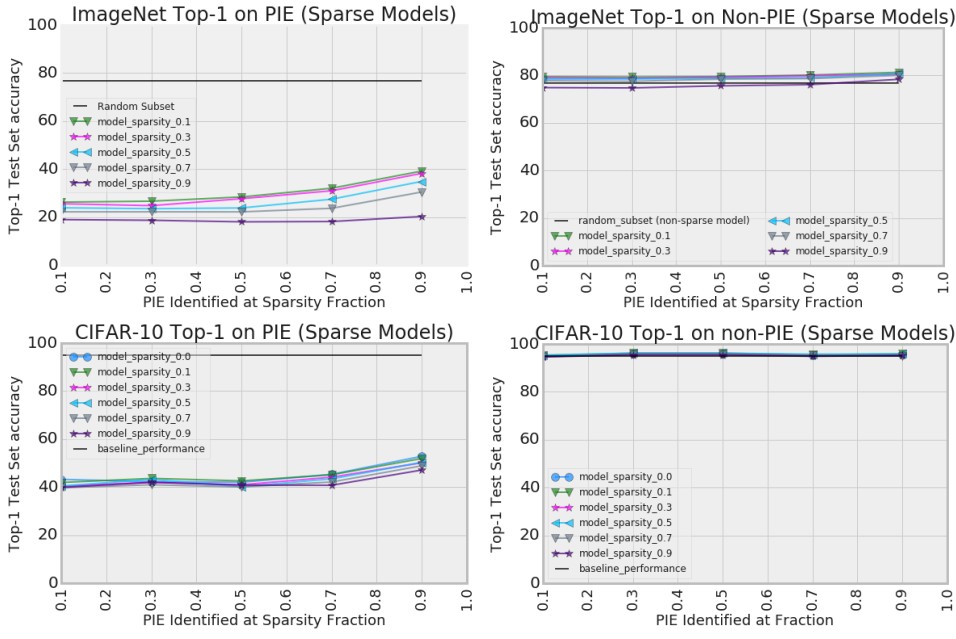

Figure 10: Excluding pruning identified exemplars (PIE) improves test-set top-1 accuracy for both ImageNet and CIFAR-10. The sensitivity to PIE images is amplified at higher levels of sparsity.

**ImageNet Robustness to ImageNet-C Corruptions (By Level of Sparsity)**

| Corruption Type | Pruning Fraction | Top-1 | Top-5 | Top-1 Relative | Top-5 Relative |
|---|---|---|---|---|---|
| brightness | 0.0 | 0.69 | 0.89 | 90.90 | 95.53 |
| brightness | 0.3 | 0.69 | 0.89 | 90.51 | 95.35 |
| brightness | 0.7 | 0.67 | 0.88 | 90.01 | 95.08 |
| brightness | 0.9 | 0.64 | 0.86 | 88.39 | 94.02 |
| contrast | 0.0 | 0.42 | 0.62 | 55.32 | 66.35 |
| contrast | 0.3 | 0.42 | 0.62 | 55.15 | 66.27 |
| contrast | 0.7 | 0.41 | 0.62 | 55.13 | 66.64 |
| contrast | 0.9 | 0.38 | 0.58 | 52.44 | 64.16 |
| defocus blur | 0.0 | 0.50 | 0.72 | 65.10 | 77.79 |
| defocus blur | 0.3 | 0.49 | 0.72 | 64.65 | 77.52 |
| defocus blur | 0.7 | 0.47 | 0.71 | 63.33 | 76.50 |
| defocus blur | 0.9 | 0.45 | 0.68 | 61.60 | 74.95 |
| elastic | 0.0 | 0.57 | 0.77 | 74.68 | 82.36 |
| elastic | 0.3 | 0.57 | 0.77 | 74.33 | 82.18 |
| elastic | 0.7 | 0.55 | 0.75 | 73.46 | 81.48 |
| elastic | 0.9 | 0.53 | 0.74 | 72.80 | 80.84 |
| fog | 0.0 | 0.56 | 0.79 | 73.52 | 85.08 |
| fog | 0.3 | 0.56 | 0.79 | 73.31 | 85.04 |
| fog | 0.7 | 0.54 | 0.78 | 72.62 | 84.68 |
| fog | 0.9 | 0.50 | 0.75 | 69.42 | 82.46 |
| gaussian noise | 0.0 | 0.45 | 0.66 | 59.42 | 70.50 |
| gaussian noise | 0.3 | 0.45 | 0.65 | 58.44 | 69.63 |
| gaussian noise | 0.7 | 0.42 | 0.62 | 56.03 | 67.52 |
| gaussian noise | 0.9 | 0.33 | 0.51 | 45.32 | 56.53 |
| impulse noise | 0.0 | 0.42 | 0.63 | 55.24 | 67.81 |
| impulse noise | 0.3 | 0.41 | 0.62 | 53.97 | 66.74 |
| impulse noise | 0.7 | 0.38 | 0.59 | 50.55 | 63.65 |
| impulse noise | 0.9 | 0.25 | 0.43 | 34.86 | 47.36 |
| jpeg compression | 0.0 | 0.66 | 0.86 | 86.00 | 92.61 |
| jpeg compression | 0.3 | 0.65 | 0.86 | 85.35 | 92.24 |
| jpeg compression | 0.7 | 0.63 | 0.85 | 84.64 | 91.78 |
| jpeg compression | 0.9 | 0.61 | 0.83 | 83.50 | 90.89 |
| pixelate | 0.0 | 0.57 | 0.78 | 75.00 | 83.80 |
| pixelate | 0.3 | 0.57 | 0.78 | 74.47 | 83.46 |
| pixelate | 0.7 | 0.55 | 0.76 | 73.25 | 82.43 |
| pixelate | 0.9 | 0.51 | 0.73 | 70.73 | 80.13 |
| shot noise | 0.0 | 0.44 | 0.64 | 57.32 | 68.78 |
| shot noise | 0.3 | 0.43 | 0.63 | 56.12 | 67.73 |
| shot noise | 0.7 | 0.40 | 0.60 | 53.19 | 64.97 |
| shot noise | 0.9 | 0.31 | 0.49 | 42.46 | 53.65 |

Table 5: Sparse models are more sensitive to image corruptions that are meaningless to a human. We measure the average Top-1 and Top-5 test set accuracy of models trained to varying levels of sparsity on the ImageNet-C test-set (the models were trained on uncorrupted ImageNet). For each corruption we consider and the relative measures see appendix (Table. 5)) we compute the average accuracy of 50 trained models across all 5 levels of corruption severity.

