# OpenReview forum: "Selective Brain Damage: Measuring the Disparate Impact of Model Pruning"
_ICLR.cc/2020/Conference — Reject_

### Official Review · AnonReviewer3 · 2019-10-16
**Official Blind Review #3**

**Rating:** 3

**Review:**

The paper claims that neural network pruning methods have different impact on accuracy in class-wise and sample-wise. To this end, the paper performs statistical tests to identify such classes (and samples) from a population of neural network models of different levels of pruning. In particular, the identified samples, called PIEs, are shown to have significantly lower test accuracy, i.e., the PIEs are much harder to classify, and a user study is performed to support this claim. The paper also demonstrates that pruned models tend to have lower accuracy against adversarial attacks or common corruptions.

In overall, the paper addresses an important problem of investigating the effects of pruning. The experiments performed here seems fairly extensive. One of my primary concerns, however, is that I am still not convinced whether the empirical findings presented here is indeed significant, as some reader might feel those results are not that surprising. I personally feel that it is much more likely that pruning gives heterogeneous effects over the classes, since current pruning schemes do not enforce the "uniform" brain damage, i.e., they simply aim to minimize total accuracy drops. Also, the accuracy differences in Table 1 do not appear to be that large in my opinion. The questions in what follows may help to resolve such concerns:

- Section 2.1: For ImageNet dataset, data imbalance in training set might be the reason of such disparate impact? As CIFAR-10 dataset is balanced, I wonder if CIFAR-10 results could be also presented in more details.
- What if the whole training and pruning is re-performed after excluding the classes which the accuracy is decreased more by pruning? Would it lead to better pruning results, or better overall accuracy?
- I feel there should be more explanation about the actual statistical implication of the use of Welch's t-test: Is it ok that S^c_t may not be independent to S^c_0? What is the key benefits of not just picking outlier classes among S^c_t with respect to c?
- What would happens if we exclude the PIEs in training set, and performs training & pruning from scratch? The overall claims may be strenghten if this could improve the pruning efficiency.
- Figure 7: I think the pure ImageNet top-1 accuracy should be also presented in each plot as a baseline for fair comparison, as this accuracy will be decreased with respect to the sparsity as well.

**Experience Assessment:**

I have published one or two papers in this area.

**Review Assessment: Checking Correctness Of Derivations And Theory:**

N/A

**Review Assessment: Checking Correctness Of Experiments:**

I carefully checked the experiments.

**Review Assessment: Thoroughness In Paper Reading:**

I read the paper at least twice and used my best judgement in assessing the paper.

---

> ### Author Response · Authors · 2019-11-08
> **Initial Discussion R3**
>
> We thank R3 for their comments and feedback. One of R3s high level concerns is that the empirical results presented are not significant or surprising. We thank R3 for this opportunity to clarify the motivation of this work. Our primary finding is that pruning incurs a non-uniform cost on a subset of classes and images. While this may be unsurprising to the reader, the implication of our findings is that a high level of caution should be exercised before deploying pruned models in sensitive domains. This is worthy of wider consideration by the academic community, as it suggests the introduction of pruning may be at odds with fairness objectives to treat certain protected attributes uniformly and/or AI safety objectives to preserve human welfare by guaranteeing a certain level of recall for certain classes.
>
> Our contribution is providing a formal framework for the academic community to evaluate the implications of pruning and appropriately internalize the trade-offs of deploying pruned model to certain domains and tasks.  For example, our methodology is salient for medical practitioners trying to evaluate how introducing sparsity biases model classification performance on different classes of skin lesions (a current real world use case of pruned models, Esteva et al. 2019).
>
> Below we respond to specific questions from R3:
>
> “the accuracy differences in Table 1 do not appear to be that large in my opinion.”  This impression may be due to a misunderstanding about the scale range. The figures in table 1 are on a scale of [0,1].  We will update these figures to be on a scale of [0%,100%] for easier readability. For example, on a [0%,100%] scale, the largest relative decrease in accuracy at 90% sparsity is for the muzzle class at 12.8%. For many sensitive tasks, this would be considered an unacceptable deterioration in performance.
>
>  In addition, we note that in table 1 we report the normalized recall difference (relative to model drop in accuracy). The per-class absolute difference in accuracy can be far larger, for example for the same muzzle class this is an absolute accuracy drop of 16.9%. We will add the absolute difference in accuracy to this table for improved readability.
>
> "Figure 7: I think the pure ImageNet top-1 accuracy should be also presented in each plot as a baseline for fair comparison, as this accuracy will be decreased with respect to the sparsity as well."
> This impression may be due to a misunderstanding, as the pure ImageNet (non-pruned) baseline is included in fig 7. The x-axis represents the level of sparsity, so the baseline model performance is plotted at x=0 (0 indicating no sparsity was introduced during training) . We will update our image captions to make this more clear to the reader.
>
> “ I wonder if CIFAR-10 results could be also presented in more details.”  We mirror that ImageNet charts produced for Cifar-10. These are included in the appendix of the current submission. We find consistent results across both ImageNet and Cifar-10 datasets (table 3,4).
>
> "What would happen if we exclude the PIEs in training set, and performs training & pruning from scratch? The overall claims may be strenghten if this could improve the pruning efficiency."
>
> This is an insightful suggestion from R3. If accepted, we would be happy to include this extension in the final version of this work.
>
> “For ImageNet dataset, data imbalance in training set might be the reason of such disparate impact”    To our knowledge, both ImageNet and Cifar-10 are treated as examples of balanced datasets by the computer vision community. Image classes in ImageNet are roughly balanced and the test set is precisely balanced for each class (https://arxiv.org/pdf/1409.0575v3.pdf).
>
> We agree with R3 it would be interesting to consider applying our methodology to a heavily unbalanced dataset (for example a language task). We would welcome such extensions of this work, as our primary contribution is to provide the formal methodology for the research community to precisely evaluate these type of trade-offs. A result that showed that under-represented classes are impacted more would likely be highly salient for domains where preserving uniform treatment for underrepresented groups is important (a large topic of interest to the fairness community).
>
> “I feel there should be more explanation about the actual statistical implication of the use of Welch's t-test: Is it ok that S^c_t may not be independent to S^c_0?”
>
> We note that S^c_t is an independent sample to S^c_0, as both  S^c_t and  S^c_0 refer to a set of models trained independently from random initialization.  Thus each set is an independent sample drawn from an underlying distribution.
>
> “What is the key benefits of not just picking outlier classes among S^c_t with respect to c?”
> The authors did not understand the question as phrased. Could R3 rephrase the question to clarify?

---

### Official Review · AnonReviewer2 · 2019-10-21
**Official Blind Review #2**

**Rating:** 3

**Review:**

This paper study how the pruning impacts the images/classes and observes which images/classes do not generalize in the pruned networks under ResNet-50 with ImageNet dataset. The authors first observe whether there exist classes exhibiting higher/lower accuracy in pruned networks compared to unpruned ones. Then, the authors define images (PIE) that generate inconsistent predictions for unpruned/pruned networks and observe the properties of PIE (e.g., whether corrupted, contains multiple labels, etc.). In particular, PIE images could be considered as `hard examples’ since even the unpruned networks misclassify them w.h.p. The authors also verify the properties of PIE and conclude the paper.

Overall, I think the paper contains some interesting observations but not enough for publication yet, due to the following reasons. First, it is well known that hard (noisy or corrupted) images are harder to be correctly classified in networks of smaller model power (Hendrycks & Dietterich, 2019). That is, results in Figure 4 and Figure 5 are not very surprising. Moreover, there is no further discussion on the hypothesis test after presenting Table 1 and Figure 3.

The paper is generally well written even it contains few typos and inconsistent presentations (e.g., legends in images are inconsistent. Both underbar and space are mixed.)


**Experience Assessment:**

I have read many papers in this area.

**Review Assessment: Checking Correctness Of Derivations And Theory:**

N/A

**Review Assessment: Checking Correctness Of Experiments:**

I carefully checked the experiments.

**Review Assessment: Thoroughness In Paper Reading:**

I read the paper at least twice and used my best judgement in assessing the paper.

---

> ### Author Response · Authors · 2019-11-13
> **Initial discussion R2**
>
> We thank R2 for their helpful feedback. We would be happy to correct for the surface level inconsistencies (legend captions and underbar, space-bar). In addition, we seek to clarify the two substantive concerns raised below.
>
> “it is well known that hard (noisy or corrupted) images are harder to be correctly classified in networks of smaller model power (Hendrycks & Dietterich, 2019). That is, results in Figure 4 and Figure 5 are not very surprising.”
>
> We thank R2 for the opportunity to clarify the contribution of our work. We note that we openly acknowledged the work of (Hendrycks & Dietterich, 2019) in our literature review -- “recent work by (Hendrycks & Dietterich, 2019; Hendrycks et al., 2019) considers the complimentary variant of this question by benchmarking ImageNet-A and ImageNet-B robustness across a limited set of dense non-sparse architectures with different numbers of parameters (for example ResNet-50 vs ResNet-101).”
>
> We believe this work is complimentary but does not devalue the contribution of this work. To our knowledge, our work is the first to consider the impact of pruning on model robustness.  We evaluate the robustness of a population of independently trained pruned models at different levels of sparsity (rather than a limited set of non-sparse models of different sizes).  Pruning is a very popular form of model compression used widely in resource constrained environments. Thus, we suggest that understanding the robustness of models is in itself a valuable contribution to the ICLR community.
>
> Varying the sparsity precisely also allows us to consider the change to robustness over a more precise range of total weights. We were able to compare a population of 30 models trained to an end sparsity t = [0.1, 0.3, 0.5, 0.7, 0.9]. In contrast, Hendrycks & Dietterich, 2019 consider one model per architecture type which have a pre-set number of weights. For example, a comparison between ResNet-18 and ResNet-50 can only provide insight into the difference in robustness for that type of architecture at 11.17 million and 24.52 million parameters respectively.
>
> Finally, we note that our contribution is not restricted to evaluating the robustness of pruned models. While these experiments are important, the bulk of the space in this paper is dedicated to introducing a formal methodology to measure the systematic impact of model pruning on certain classes (section 2.1, 3.1, p.2-5) and individual exemplars (section 2.2, 3.2, p 4-7).
>
> “Moreover, there is no further discussion on the hypothesis test after presenting Table 1 and Figure 3. “
>
> Would R2 be able to provide guidance as to what additional treatment would be helpful within the space constraints of the submission? We note that while space constraints prevented the introduction of cifar results within the body of the manuscript, we do include these in the appendix as well as additional ImageNet results at different levels of sparsity.
>
> In addition, our current submission dedicates several pages to the introduction and discussion of the hypothesis test results. In section 2.1 (p. 2-4), we introduce a statistical framework for measuring the systematic impact of pruning different classes. In section 3.1 (p.5),  we discuss the results of the normalized recall difference in) where we reference the charts in table 1 and fig.3. We would be happy to consider suggestions from R2 about areas for additional discussion or clarification.

---

> > ### Comment · AnonReviewer2 · 2019-11-13
> > **Thank you for your response**
> >
> > I've read your response. As the authors and other reviewers mentioned, I acknowledge that the problem aimed at this paper is important and the authors made the first observation on the robustness of pruned networks. However, I still believe that the contribution of the paper is limited due to the following reason.
> >
> > -The robustness of pruned networks is an important issue to resolve, but the observation that ``pruned model is not uniformly robust for all classes'' itself is not quite surprising as I mentioned in the initial review.
> >
> > -Meanwhile, the authors only provide observations but did not clearly explain why such observations occur (e.g., properties of robust classes under pruned models) and/or how to resolve this issue (e.g., developing methods for obtaining uniformly robust pruned models). In particular, I would like to recommend to provide such an explanation / a solution for "Would R2 be able to provide guidance as to what additional treatment would be helpful within the space constraints of the submission?".

---

### Official Review · AnonReviewer1 · 2019-10-24
**Official Blind Review #1**

**Rating:** 3

**Review:**

This paper presents an empirical study on the effect of pruning to the model performance on each class and example, which leads to a novel finding that it has disparate effects to each sample. Specifically, the authors have found out that examples that are affected the most by pruning are more difficult to classify even for the non-pruned network, due to low image quality, mislabeling, or being atypical from the class prototype, and performed a further human study to analyze the source of difficulty. Moreover, the authors performed an additional experiment, which shows that the sparse models are brittle against natural adversarial examples.

Pros
- The paper provides a novel insight on the effect of pruning at the class and the example level, which could lead to a more effective pruning approach that exploit this findings.

Cons

- The paper only provides a novel finding but not the solution on how to tackle this problem, and thus the paper looks incomplete. After section 3.3, I was expecting to see some approaches to tackle this problem but the paper abruptly ended.

- The effect of pruning could largely differ from one method to another, but the authors do not experimentally compare the effects of different pruning methods. Also, it is highly likely that the findings discussed in the paper may be only true for input-independent pruning approaches, and may not generalize to input-dependent pruning method. The authors need to perform extensive study of both input-dependent and input-independent pruning approaches to validate their points.

In sum, the paper provides a novel insight on how pruning affects the performance at the example level, but does not provide a solution, and the current set of experiments is insufficient to validate that the empirical findings that the authors report generalize to other types of pruning approaches, such as input-dependent pruning. Thus I believe that the paper is not ready for publication yet, and vote for rejecting this paper in its current form.


**Experience Assessment:**

I have published in this field for several years.

**Review Assessment: Checking Correctness Of Derivations And Theory:**

I assessed the sensibility of the derivations and theory.

**Review Assessment: Checking Correctness Of Experiments:**

I carefully checked the experiments.

**Review Assessment: Thoroughness In Paper Reading:**

I read the paper thoroughly.

---

> ### Author Response · Authors · 2019-11-12
> **Discussion about motivation of paper and scope of contributions**
>
> We thank R1 for their feedback, and acknowledge that we do not propose a new method to solve the non-uniform impact of model pruning on different classes and images. We believe our contribution is none the less of value to the ICLR community, because to our knowledge we are the first work to propose a formal framework for measuring the per-class and image impact of pruning deep neural networks. We consider an unexplored and timely question given the widespread use of pruning as a compression technique in many sensitive domains.
>
> Our conclusion, that a small subset of classes and images bear the brunt of pruning, has wide ranging implications for the use of pruned models in sensitive domains. The introduction of pruning may be at odds with fairness objectives to treat certain protected attributes uniformly and/or AI safety objectives to preserve human welfare when predictions by guaranteeing a certain level of recall for certain classes.
>
> Our contribution illustrates that top-1 accuracy alone tells a painfully incomplete picture, and that for these sensitive domains additional measures of model generalization are certainly needed. We propose such formal measures of generalization difference in this manuscript. We will update our manuscript to make this motivation and the scope of our contributions more clear.
>
> "The effect of pruning could largely differ from one method to another, but the authors do not experimentally compare the effects of different pruning methods."
>
> R1 is correct that the results that we report (while consistent across datasets) are only for a single pruning method. We were restricted in our choice of pruning method by the need to pre-specify the final level of sparsity to allow for clean comparison across trained models. We also note that magnitude pruning is a widely used (open source and integrated into the tensorflow library) and considered to be state of art (Gale et al. 2019).
>
> We agree that a consideration of additional methods would be of value, and have committed to open sourcing our code so that we are not the bottleneck for further reproducibility. We note that our methodology is pruning method agnostic, and can easily be extended to other method.
>
> We wanted to seek clarification on what R1 means by input-independent pruning approaches? Can R1 reference an open source implementation of such an approach or prior work to help clarify the use of this terminology.

---

### Decision · Program_Chairs · 2019-12-19

**Decision:**

Reject

**Comment:**

This work investigates neural network pruning through the lens of its influence over specific exemplars (which are found to often be lower quality or mislabelled images) and how removing them greatly helps metrics.
The insight from the paper is interesting, as recognized by reviewers. However, experiments do not suggest that the findings shown in the paper would generalize to more pruning methods. Nor do the authors give directions for tackling the "hard exemplar" problem. Authors' response did provide justifications and clarifications, however the core of the concern remains.
Therefore, we recommend rejection.